# Prediction of the Postoperative Outcome in Liver Resection Using Perioperative Serum Lactate Levels

**DOI:** 10.3390/jcm12062100

**Published:** 2023-03-07

**Authors:** Sebastian Recknagel, Sebastian Rademacher, Claudia Höhne, Andri A. Lederer, Undine G. Lange, Toni Herta, Daniel Seehofer, Robert Sucher, Uwe Scheuermann

**Affiliations:** 1Department of Visceral, Transplantation, Vascular and Thoracic Surgery, University Hospital of Leipzig, 04103 Leipzig, Germany; 2Department of Anesthesiology, Pain Therapy, Intensive Care and Emergency Medicine, DRK Hospital Berlin-Koepenick, 12559 Berlin, Germany; 3Department of Gastroenterology and Oncology, Division of Hepatology, University Hospital Leipzig, 04103 Leipzig, Germany

**Keywords:** liver resection, lactate, prognostic factor, outcome, Clavien–Dindo classification, complication, morbidity, mortality

## Abstract

Background: The aim of our study was to analyze perioperative lactate levels and their predictive value for postoperative mortality and morbidity after liver resection. Methods: The clinicopathological characteristics and outcomes of 152 patients who underwent liver resection for benign and malign diagnoses were analyzed retrospectively. Lactate concentrations at three different time points, (1) before liver resection (LAC-PRE), (2) after liver resection on day 0 (LAC-POST), and (3) on day one after the operation (LAC-POD1) were assessed regarding the prognostic value in predicting postoperative complications and mortality according to the Clavien–Dindo (CD) classification. Results: The rates of postoperative complications (CD ≥ IIIb) and mortality rates were 19.7% (N = 30) and 4.6% (N = 7), respectively. The LAC-PRE levels showed no correlation with the postoperative outcome. The ROC curve analysis showed that LCT-POST and LCT-POD1 values were moderately strong in predicting postoperative morbidity (0.681 and 0.768, respectively) and had strong predictive accuracies regarding postoperative mortality (0.800 and 0.838, respectively). The multivariate analysis revealed LAC-POST as a significant predictor of postoperative complications (CD ≥ IIIb: OR 9.28; 95% CI: 2.88–29.9; *p* < 0.001) and mortality (OR 11.69; 95% CI: 1.76–77.7; *p* = 0.011). Conclusion: Early postoperative lactate levels are a useful and easily practicable predictor of postoperative morbidity and mortality in patients after liver resection.

## 1. Introduction

In recent decades, the number of liver surgeries for the treatment of primary and secondary liver tumors has been steadily increasing. However, despite technical improvements in surgical techniques and anesthesiologic care, the incidence of major complications after liver resection (LR) still ranges from 19 to 23% [1,2,3], and mortality rates are approximately 6% [3,4,5,6,7]. Surgical complications such as bile leaks or intraabdominal infection and non-surgical complications such as hepatic or renal dysfunction, respiratory failure, and sepsis impact the outcome and mortality. Therefore, numerous studies have attempted to find simple parameters (i.e., hepatic steatosis, patient age, serum bilirubin) and scores to predict the patient outcome after LR [1,7,8,9,10,11]. With a suitable marker, clinicians could estimate the individual risk of postoperative complications of each patient after LR and set up an optimal patient management protocol.

In this context, the amount of lactate seems to be a potential parameter that can be determined quickly and easily in everyday clinical practice. Lactate is a byproduct of anaerobic cellular metabolism and a marker for reduced organ perfusion and end-organ damage; hence, it can be used to guide fluid therapy and to detect critically ill patients with hypoperfusion [12,13,14]. In the human body, the liver is responsible for most of the lactate clearance [15] and thus, the quality of liver parenchyma, the extent of LR, anaerobic time during vascular occlusion and surgical resection, as well as intraoperative blood loss, accompanied by hypoperfusion of the liver and hypoxia; all have a direct impact on the lactate level [16,17,18,19,20].

To date, only a few studies have investigated the role of perioperative lactate levels as predictors for the postoperative outcome after LR [17,18,19,20,21,22,23,24,25,26], and differences in study design and a lack of data regarding early postoperative lactate release make further studies necessary.

Therefore, the aim of the study was to compare lactate levels at different time points in patients undergoing various extents of LR and to determine their association with postoperative complications and mortality.

## 2. Patients and Methods

### 2.1. Data Collection and Study Population

Medical data from all adult patients (≥18 years of age) who underwent LR at the University Hospital of Leipzig between April 2016 and September 2017 were retrospectively analyzed. Patients with missing data (N = 2) or an unknown time of lactate measurement (N = 2) were excluded from the study. Data were collected from anesthesiologic and operative documentation of the patient’s history, the intraoperative procedure, histological results, the process at the intensive care unit, and the medical discharge report.

Before surgery, each case was reviewed in a multidisciplinary tumor-board meeting. Patients were not considered for surgery in case of advanced liver disease (e.g., liver cirrhosis Child–Pugh stage B or C, extensive portal vein thrombosis), poor general conditions, severe cardial or pulmonal failure. Sufficient liver parenchyma and liver function were measured by CT volumetry and the LiMAx^®^ test [27]. If the future liver remnant was too small, patients received portal vein embolization (PVE) in order to obtain resectabilty. About one month after PVE, CT volumetry was repeated to ensure a sufficient volume increase in the future liver remnant. Prior to a major liver resection, 25–30% of the total liver volume or a future remnant liver volume of 0.8% of the patient’s body weight was considered sufficient. Furthermore, the future residual liver function—calculated from the preoperative LiMAx^®^ test and future remnant liver volume—should be above 100 µg/h/kg [27].

The medical data analysis comprised patient demographics (age, gender, and body mass index [BMI, weight in kg/height in m^2^]), preoperative performance status according to the ASA (American Society of Anesthesiology) classes I to III, and intraoperative surgical data (operation time, blood loss, transfusion requirement [defined as units of red blood cells, albumin, or fresh frozen plasma substitution], Pringle maneuver, bile duct drain [T-drain] placement, the extent of resection, and lymphadenectomy), histopathological findings and post-operative data (length of hospital stay, postoperative complications during the hospital stay). The extent of resection was described according to the Brisbane classification [28] as being a major (exceeding three segments) or minor (less than four segments or atypical resections) LR. Accordingly, major LRs included right or left hemihepatectomy or right or left trisectorectomy (extended hemihepatectomies). In addition to these classic approaches to resection, a considerable number of patients underwent additional procedures such as vascular or biliary resection and resection of adjacent structures. These approaches were classified as “complex resection”. Lymphatic dissection during surgery always included the lymph nodes of the hepatoduodenal ligament and those at the common hepatic artery. In very few cases the retropancreatic and celiac lymph nodes were also excised.

During open liver resection (OLR), parenchymal transection was performed with the Cavitron Ultrasonic Surgical Aspirator (CUSA) and while smaller vascular and biliary structures were divided between titanium clips, larger structures were ligated. During laparoscopic liver resection (LLR), ultrasonic shears (Harmonic ACE, Ethicon^®^) were used as a mainstay of tissue dissection and parenchymal transection. A laparoscopic CUSA was only used during right or extended right hemi-hepatectomies for exposure of the middle hepatic vein and its tributaries and for intrahepatic exposure of the right bile duct. Additional hemostasis was performed by bipolar forceps and irrigation. In specific cases, such as resections of superior segments 7–8, a hand-assisted laparoscopic approach was applied. For the vascular inflow control, a tourniquet was placed around the hepatoduodenal ligament and temporarily closed during parenchymal resection when necessary (Pringle maneuver). All patients with LRs received at least overnight intensive care and were transferred to the normal ward at the earliest on the day after surgery.

The study was approved by the ethics committee of the University of Leipzig, number 142/18-EK, and the study protocol was performed in accordance with the relevant guidelines. Informed consent was waived by the ethics committee of the University of Leipzig due to the retrospective nature of the study.

### 2.2. Outcome Measures

Post-operative complications occurring in the first three months after LR were analyzed. The complications included delayed wound healing, wound infection, bleeding, development of hematoma and lymphoceles, pneumonia, unplanned intubation, pulmonary embolism, >48 h ventilator requirement, pneumothorax, renal failure, postoperative liver failure, ascites, bile leakage, stroke or cerebral vascular accident, cardiac arrest, myocardial infarction, deep venous thrombosis, or systematic sepsis. Post-hepatectomy liver failure was defined as a deterioration of the liver functions indicated by laboratory values, such as an increased INR and concomitant hyperbilirubinemia on or after postoperative day five [29], or clinical symptoms such as hepatic encephalopathy. The Clavien–Dindo (CD) classification was used for grading postoperative complications [30]. Major morbidity was defined as being CD IIIb or greater (CD ≥ IIIb), which requires surgical, endoscopic, or radiologic intervention under general anesthesia. Mortality was defined as any death occurring within 90 days following the date of surgery or in-hospital mortality.

The lactate concentrations were determined from blood samples taken before liver resection (LAC-PRE), after liver resection on day 0 (LAC-POST), and on day one day after the operation (LAC-POD1). The limit value for lactate is <2.0 mmol/L. Other laboratory values included in the analysis are preoperative bilirubin, base excess, and creatinine.

### 2.3. Statistical Analysis

For comparison between the groups, the appropriate statistical significance test, including the Student’s *t*-test, the chi-squared test, analysis of variance (ANOVA), the Kruskal–Wallis test, and the Wilcoxon–Mann–Whitney test were used. Receiver operator characteristic (ROC) curves were generated to determine the optimal diagnostic criterion threshold for predicting postoperative outcomes. A ROC curve displayed the false positive rate on the x-axis (specificity), and the true positive rate on the y-axis (sensitivity) for varying test thresholds, thereby plotting the performance of a diagnostic test. The predictive accuracy was measured by the area under the ROC curve (AUC) whereby higher AUC values represent greater accuracy. An AUC of 1.0 represents perfect discrimination (perfect sensitivity and specificity), whereas an AUC of 0.5 represents an essentially worthless test. The cut-off values for continuous variables were chosen by receiver operator characteristic (ROC) curves analysis and calculation of the Youden index according to each point (Youden’s J statistic: J = sensitivity + specificity − 1) [31]. These values were used for the subsequent uni- and multivariate analyses. Uni- and multivariate logistic regression analyses were used to evaluate the association between independent variables and binary postoperative outcomes (CD ≥ IIIb and mortality). For the multivariate analyses, we used a forward stepwise regression model including only clinically relevant variables and those presenting *p* < 0.05 in univariate analysis.

SPSS software, version 21.0 (SPSS Inc., Chicago, IL, USA) and GraphPad Prism software, version 9.4.1 (GraphPad Software, San Diego, CA, USA) were used for statistical analysis and graphs. A *p* value < 0.05 was considered statistically significant. Unless otherwise indicated, the baseline data are presented as median values with the standard deviation (SD).

## 3. Results

### 3.1. Baseline Characteristics

The overall characteristics of the patients and their surgeries are summarized in Table 1. The indications for surgery were 116 malignant (76.3%) and 36 benign (23.7%) tumors. Liver resection (LR) was performed by 107 open liver resections (OLR) (70.4%), 36 laparoscopic liver resections (LLR) (23.7%), and nine hand-assisted laparoscopic liver resections (HALLR) (5.9%). Ten patients received portal vein embolization prior to extended hemihepatectomy. In one case an in situ split with liver segment resection was performed followed by a left hemihepatectomy eight days later. Sixty-eight patients (56.6%) received surgery on the upper abdomen prior to LR, and in seven cases (4.6%) a conversion from LLR to OLR was necessary. The operative procedures included 102 minor (67.1%) and 50 major (32.9%) LRs. Due to tumor invasion, the following extrahepatic resections were necessary: hemicolectomy (N = 2), partial diaphragm (N = 6), colon section (N = 2), extrahepatic bile duct (N = 14), adrenal gland (N = 1), partial kidney (N = 1), portal vein bifurcation (N = 3), and atypical partial lung (N = 3) resection.

The median hospital stay was 9.0 ± 12.2 days (range: 3–72 days). The overall postoperative complications rate was 38.8% (N = 59) and the overall mortality rate was 4.6% (N = 7). The summary of the postoperative outcomes is shown in Table 2. All deaths occurred after extended hemihepatectomies within the first months after LR (range: 3–20 days). Five of these patients received a biliodigestive anastomosis. The causes of death included septic shock and multiple organ failure (N = 4), liver insufficiency (N = 1), bowel ischemia (N = 1), and pulmonary embolism and cardiac arrest (N = 1), respectively.

### 3.2. Correlation of Lactate Levels with Postoperative Outcomes

The distribution of lactate levels in the study populations before and after resection as well as on the first postoperative day is shown in Figure 1. On day 0 after LR (LAC-POST, 1.8 ± 1.6 mmol/L) and one day after the surgery (LAC-POD1,1.8 ± 2.5 mmol/L), the average lactate levels showed significant differences compared to the lactate concentrations before liver resection (LAC-PRE, 0.8 ± 0.5, mmol/L) (*p* < 0.001 each), whereas LAC-POST and LAC-POD1 were comparable (*p* = 0.310). One day after surgery, the lactate concentration was still elevated (≥2 mmol/L) in 41% of the patients.

The average postoperative lactate concentrations (LAC-POST, LAC-POD1) were significantly increased in patients suffering from postoperative complications (Table 3).

Apart from the postoperative outcome, the lactate levels were also associated with intraoperative characteristics. There was a significant positive correlation between intraoperative blood loss, operation time, and both postoperative lactate levels (*p* < 0.001 each). A prolonged Pringle maneuver was also significantly associated with elevated LAC-POST (*p* = 0.008), but not with LAC-POD1 (*p* = 0.092). Postoperative lactate levels correlate with the extent of LR. Postoperative lactate levels were significantly increased in patients after extended hemihepatectomies (trisectorectomies) when compared with patients after minor LR (*p* < 0.001 each). The relationship between the intraoperative blood loss, duration of the Pringle maneuver, operation time, extent of liver resection, and lactate concentration after LRs is shown in Figure 2.

The prognostic values of the study parameters in predicting severe postoperative complications (CD ≥ IIIb) and mortality were assessed using ROC curve analysis and probability determination (Figure 3A,B). In our cohort, the LAC-PRE levels had no predictive power regarding morbidity or mortality after LR. However, the AUC values of LAC-POST and LAC-POD1 were moderately strong in predicting postoperative morbidity (0.681, and 0.768, respectively) whereby the corresponding cut-off values were 2.8 and 2.4 mmol/L. The AUC values of LAC-POST and LAC-POD1 regarding postoperative mortality indicated strong predictive accuracies (0.800, and 0.838, respectively) and the corresponding cut-off values were 3.1 and 5.4 mmol/L, respectively.

The predictors of postoperative morbidity and mortality after LR are shown in Table 4 and Table 5, respectively. The cut-offs obtained from the ROC analysis were used to divide patients into two groups for uni- and multivariate logistic regression analysis (MVA). In the univariate analysis, LAC-POST and LAC-POD1 were significantly associated with postoperative morbidity (CD ≥ IIIb) and mortality (*p* < 0.001/0.001 and *p* = 0.009/0.021). After adjusting for co-variables in MVA, the patient’s age, complex resections (e.g., biliodigestive anastomosis or vascular reconstruction), duration of the Pringle maneuver, and LAC-POST (OR 9.28; 95% CI: 2.88–29.9; *p* < 0.001) were significant predictors for severe postoperative complications (CD ≥ IIIb). Regarding postoperative mortality, MVA revealed complex resection and LAC-POST (OR 11.69; 95% CI: 1.76–77.7; *p* = 0.011) as independent risk factors whereby LAC-POD1 failed to be identified as an independent predictor of morbidity and mortality in MVA. However, calculation and analysis of lactate differences one day after LR showed advantages regarding the prediction of postoperative complications, especially mortality (OR 20.72; 95% CI: 2.72–157.81; *p* = 0.003).

## 4. Discussion

Based on the results, the predictive value of perioperative lactate levels on the short-term outcomes after liver resection is discussed here.

### 4.1. Preoperative Lactate Level

In the current study, preoperative lactate levels failed to predict the outcome after LR. In reviewing the literature, only two publications were found on this specific topic. In a recent study by Popescu et al., which investigated the rates of postoperative liver failure in a group of 55 patients after major LR, the lactate levels prior to the operation failed to predict its outcome [24]. In contrast, a retrospective study by Riediger et al. including 337 patients undergoing OLR demonstrated preoperative lactate levels as independent risk factors for in-hospital death (OR: 1.474; *p* = 0.004) [25]. The divergent results of these two studies may be explained by their different study populations, as hepatic lactate metabolism is determined by the quality of liver parenchyma and patient comorbidities, among other factors. Whereas Riediger et al. included a heterogeneous group of patients, who could have various causes of increased preoperative lactate levels, such as liver trauma, hepatitis, or liver cirrhosis, Popescu et al. excluded those patients with a potentially worse postoperative outcome [24,25]. In our cohort, all operations were elective. The patients were preselected due to routine clinical evaluation (including LiMAx^®^ [maximum liver function capacity] test), and their lactate levels were mainly determined by intra- and postoperative procedures.

### 4.2. Early Postoperative Lactate Level (Day 0)

Lactate levels measured directly after surgery correlate with morbidity and mortality after LR [17,19,20,22,23,24,26]. Our results are in line with previous studies, which revealed lactate levels at the end of surgery as being an independent risk factor for postoperative complications and death [17,19,26]. In the current study, the lactate cut-off values are almost identical to the results of the multi-center analysis by Vibert et al., who defined cut-off values of 2.8 mmol/L and 3.0 mmol/L for the prediction of severe morbidity and 90-day mortality after LR, respectively [19]. These results may be relevant for surgeons and intensive care physicians in their daily clinical practice. They support the proposal by Wiggans et al., who postulated that lactate levels could be used to guide clinicians regarding the requirement of intensive care after LR, as patients with a normal postoperative lactate level are unlikely to suffer from hepatic dysfunction [20]. Furthermore, in the intensive care unit, the early postoperative lactate could be used to optimize hemodynamics and fluid therapy [23].

In our study, surgery time, duration of the Pringle maneuver, intraoperative blood loss, and extent of LR correlate with elevated postoperative lactate levels although none of the parameters were independently associated with postoperative morbidity or mortality. Previous studies showed several pre- and intraoperative factors to be associated with postoperative hyperlactatemia [17,19,20], and especially patients with diabetes display elevated lactate levels compared to healthy subjects, which most likely is due to the impairment of gluconeogenesis [19,20,32]. Unfortunately, in our cohort, data regarding patients’ pretreatment and comorbidities, such as the number and type of chemotherapies, diabetic medication, and other specific drugs associated with increased lactate levels, are incomplete, and conclusions about their influence on postoperative outcome cannot be drawn.

### 4.3. Lactate Clearance (Day 1)

In general, reduced lactate clearance indicates restricted microcirculation [33] and decreased liver function since up to 70% of the lactate in the human body is eliminated by the liver [15]. Especially in critically ill patients such as those suffering from sepsis, lactate clearance is a strong predictor of mortality [13,34]. In reviewing the literature, only one previous study was found, which analyzed the association between lactate clearance and the early postoperative outcome after LR. In their retrospective study published in 2015, Pagano et al. analyzed lactate clearance on day five after LR (defined as lactate at postoperative day five minus lactate at ICU presentation), without showing any significance in multivariate analysis [23]. However, the study population consisted of only 45 patients undergoing exclusively extended hepatectomies. In our study cohort, the lactate level one day after LR also failed to be an independent predictor of morbidity and mortality although the lactate clearance within the first postoperative day was a strong prognostic factor of patient survival. All deaths in our study population occurred after extended resections, which can lead to transient or permanent hepatic insufficiency and impairment of the lactate metabolism. It can be assumed that hyperlactatemia is not a primary manifestation of increased lactate production, as is the case in septic patients, but rather a result of decreased liver function and lactate clearance. Therefore, the determination of early lactate clearance as an indicator of reduced liver function seems to have a potential benefit in patients after major liver resection.

### 4.4. Postoperative Mortality

Our overall morbidity and mortality rates are comparable to those of other high-volume centers. In our study, the mortality rate was 22.6% for patients after extended hemihepatectomies (7 out of 31 patients), which is in line with previous reports. In a retrospective analysis by Filmann et al. including about 110,000 liver procedures performed in Germany between 2010 and 2015, the average hospital mortality rate after major liver resection was 10.4%. In addition, extended hemihepatectomies with biliodigestive anastomosis even led to a hospital mortality rate of 25.5% [5]. Besides postoperative hepatic insufficiency, bile leakage after biliary reconstruction contributes to the high postoperative complication rate after major liver resections [35].

### 4.5. Limitations

There are some notable limitations of this study that should be mentioned, the first of which is the relatively small number of patients that were included and its retrospective design. Second, the data concerning the lactate-contributing factors during anesthesia in the operating room and the intensive care unit are incomplete. Further studies with the measurement of factors such as the administration of catecholamines and vasoactive drugs, drugs associated with increased lactate levels, and circulatory parameters would be of interest.

## 5. Conclusions

Our study presents early postoperative lactate levels and clearance as a predictor of the postoperative outcome after liver resections. Furthermore, the correlation was independent of the diagnosis, the complexity of the resection, and the resection technique. Patients with elevated lactate should be monitored carefully due to the higher risk of morbidity and mortality. Further prospective studies are required to confirm our results and to determine the optimal postoperative care and therapy for these patients.

## Figures and Tables

**Figure 1 jcm-12-02100-f001:**
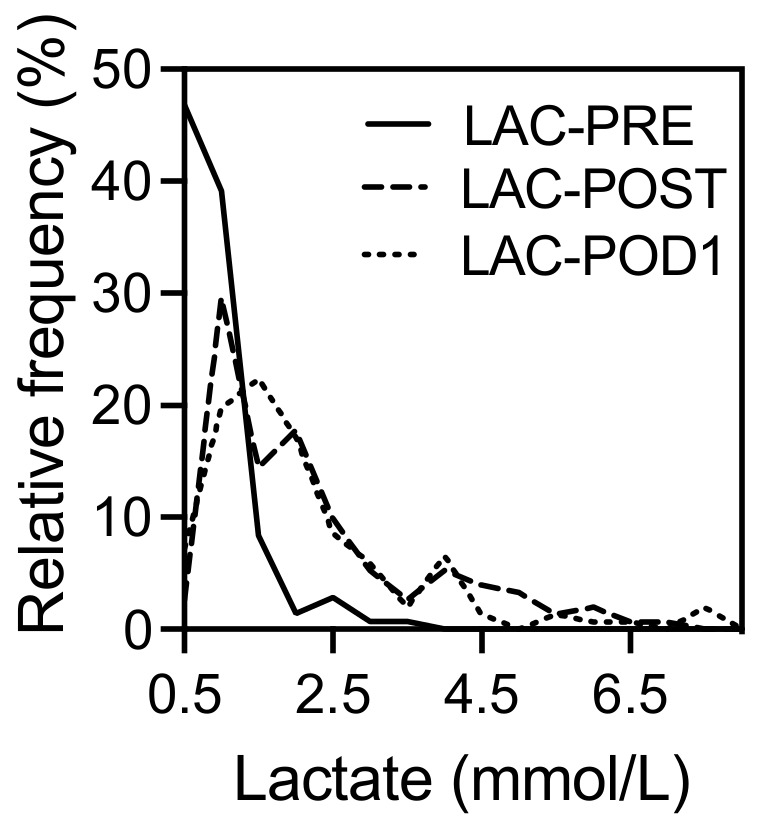
Perioperative distribution of lactate concentration in 152 patients who underwent liver resection. LAC-PRE, lactate level before liver resection; LAC-POST, lactate after liver resection on day 0; LAC-POD1, lactate on day one after liver resection.

**Figure 2 jcm-12-02100-f002:**
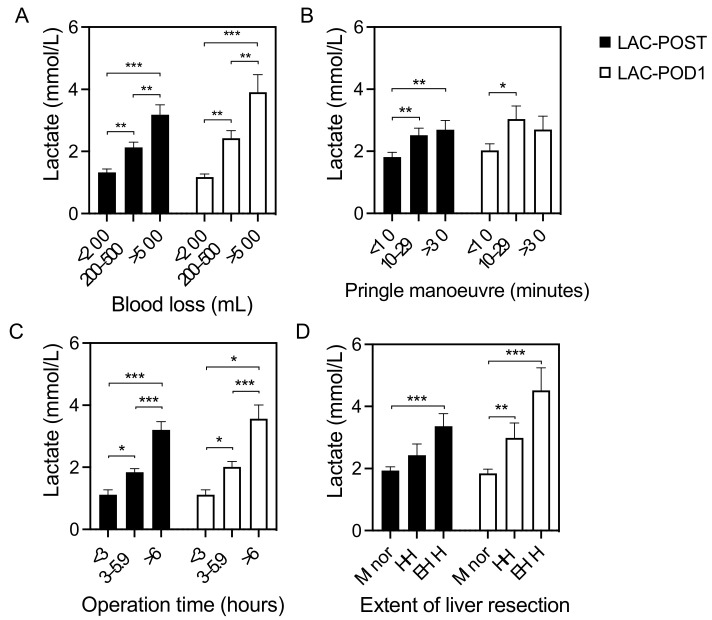
Variables associated with postoperative lactate concentration. Relationship between the (**A**) intraoperative blood loss, (**B**) duration of the Pringle maneuver, (**C**) operation time, and (**D**) extent of liver resection and lactate concentration after liver resection. HH, hemihepatectomy; EHH, extended hemihepatectomy; LAC-POST, lactate after liver resection on day 0; LAC-POD1, lactate on day one after liver resection; minor, minor liver resection. Data are shown as mean values ± Standard error of the mean. * *p* < 0.05, ** *p* < 0.01, *** *p* < 0.001.

**Figure 3 jcm-12-02100-f003:**
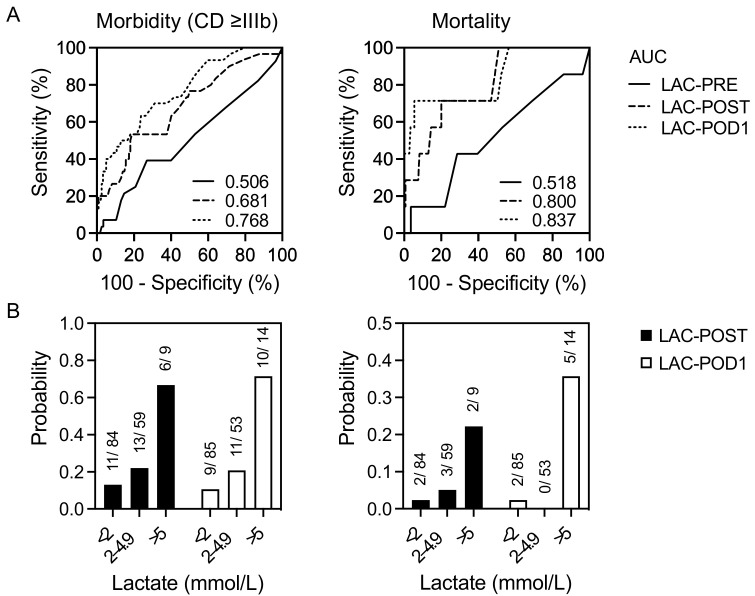
ROC curves and probability of perioperative lactate levels in predicting postoperative complications and mortality in 152 patients undergoing liver resection. Receiver operator characteristic (ROC) curves (**A**) for lactate as discriminators of the increased rate of postoperative morbidity and mortality. Probability (**B**) of postoperative morbidity and mortality according to lactate concentration. CD ≥ IIIb, Clavien–Dindo ≥ IIIb; LAC-POST, lactate after liver resection on day 0; LAC-POD1, lactate on day one after liver resection.

**Table 1 jcm-12-02100-t001:** Baseline characteristics of the overall study population.

Variables	N = 152
Patient Characteristics	
Mean age, years	60.5 ± 15.4
Gender, male/female, n (%)	86 (56.6)/66 (43.4)
BMI, kg/m^2^	26.1 ± 4.7
ASA score, I/II/III, n (%)	6 (3.9)/74 (48.7)/72 (47.4)
Parenchymal liver disease, n (%)	
Steatosis hepatis/fibrosis/cirrhosis	21 (13.8)/6 (3.9)/20 (13.2)
Diagnosis, n (%)	
Benign/malign	36 (23.7)/116 (76.3)
Primary malign hepatic tumors, HCC/CCA/others	30 (19.7)/22 (14.5)/6 (3.9)
Hepatic metastases, CRLM/nCRLM	44 (28.9)/14 (9.2)
Benign lesions, benign tumors/others	27 (17.8)/9 (5.9)
Preoperative laboratory parameters	
Base excess, mmol/L	−1.0 ± 2.9
Lactate, mmol/L	0.8 ± 0.5
Bilirubine, µmol/L	7.7 ± 15.4
Creatinine, µmol/L	73 ± 27.3
Portal vein embolization/in-situ split	10 (6.6)/1 (0.7)
Surgery	
OLR/HALLR/LLR, n (%)	107 (70.4)/9 (5.9)/36 (23.7)
Minor/major resection, n (%)	102 (67.1)/50 (32.9)
Type of resection, n (%)	
Atypical	20 (13.2)
Segmentectomy, −/+ atypical resection	41 (27.0)/27 (17.8)
Bisegmentectomy −/+ atypical resection	6 (3.9)/8 (5.3)
Hemihepatectomy left/right	10 (6.6)/9 (5.9)
Extended hemihepatectomy left/right	9 (5.9)/22 (14.5)
Complex resection, n (%)	28 (18.4)
Biliodigestive anastomosis, n (%)	14 (9.2)
Lymphadenectomy, n (%)	56 (36.8)
T-tube, n (%)	46 (30.3)
Pringle manoeuvre (PM), n (%)	93 (61.2)
PM duration, minutes	17 ± 16.3
Intraoperative blood loss, mL	335 ± 445
Transfusion, n (%)	
RBC/FFP/albumin	6 (3.9)/8 (5.3)/10 (6.6)
Operation time, minutes	312 ± 118

ASA, American Society of Anesthesiology; BE, base excess; BMI, body mass index; CCA, cholangiocarcinoma; CRLM, colorectal liver metastases; FFP, fresh frozen plasma; HALLR, hand-assisted laparoscopic liver resection; HCC, hepatocellular carcinoma; LLR, laparoscopic liver resection; nCRLM, non-colorectal liver metastases; OLR, open liver resection; PM, Pringle maneuver; POD1, postoperative day 1; POST, postoperative; PRE, preoperative; RBC, red blood cells. Data are shown as median values ± SD.

**Table 2 jcm-12-02100-t002:** Summary of the postoperative outcome in 152 patients after liver resection.

Variables	N = 152
ICU	
ICU stay, days	1.0 ± 2.7
Mechanical ventilation, n (%)	20 (13.2)
Transfusion, n (%)	
RBC/FFP/albumin	12 (7.9)/3 (2.0)/5 (3.3)
Morbidity, n (%)	
Overall complications	59 (38.8)
Total events	116
Systematic sepsis	5 (3.3)
Liver/biliary	
Liver failure/bile leakage/ascites/liver abscess/cholangitis	7 (4.6)/7 (4.6)/4 (2.6)/3 (2.0)/3 (2.0)
Pulmonary	
Pleural effusion/pneumonia/unplanned intubation/ventilator requirement > 48 h/pulmonary embolism/pneumothorax	12 (7.9)/2 (1.3)/4 (2.6)/2 (1.3)/4 (2.6)/3 (2.0)
Genitourinary	
Renal failure	8 (5.3)
Wounds	
Delayed wound healing/fascial dehiscence/wound infection	26 (17.1)/7 (4.6)/5 (3.3)
Vascular	
Bleeding/hematoma/embolism liver arteria/deep venous thrombosis	5 (3.3)/2 (1.3)/1 (0.7)/1 (0.7)
Other	
Cardiac arrest/cerebral vascular accident/intestinal ischemia/lymphoceles	1 (0.7)/1 (0.7)/1 (0.7)/2 (1.3)
Mortality, n (%)	7 (4.6)
Clavien–Dindo I/II/IIIa/IIIb/IV/V	18 (11.8)/4 (2.6)/7 (4.6)/14 (9.2)/9 (5.9)/7 (4.6)
Hospital stay, days	9.0 ± 12.2

FFP, fresh frozen plasma; ICU, intensive care unit; RBC, red blood cells.

**Table 3 jcm-12-02100-t003:** Comparison of lactate levels and postoperative morbidity and mortality in 152 patients who underwent liver resection. of the postoperative outcome in 152 patients after liver resection.

Variable	Morbidity (CD ≥ IIIb)		*p*-Value	Mortality		*p*-Value
	No	Yes		No	Yes	
Lactate, mmol/L						
LAC-PRE	0.8 ± 0.5	0.8 ± 0.5	0.809	0.8 ± 0.5	0.8 ± 0.7	0.638
LAC-POST	1.6 ± 1.2	2.9 ± 2.4	<0.001	1.8 ± 1.4	4.0 ± 3.4	<0.001
LAC-POD1	1.6 ± 1.5	3.0 ± 4.1	<0.001	1.7 ± 1.8	7.6 ± 5.9	<0.001

Data are shown as median values ± SD. POD1, postoperative day 1; POST, postoperative day 0, PRE, preoperative.

**Table 4 jcm-12-02100-t004:** Predictors of postoperative morbidity in 152 patients who underwent liver resection.

Variables	Morbidity (Clavien-Dindo ≥ IIIb)								
	ROC					UVA	Multivariate Analysis		
	AUC	Cut-Off Value	Sen (%)	Spe (%)	LRa	*p*-Value	OR	95% CI	*p*-Value
Patient characteristic									
Age	0.676	64.9 years	73.3	67.2	2.24	<0.001	7.604	2.474–23.370	<0.001
ASA score I–II vs. III						0.007	NS	NS	NS
Laboratory tests									
Bilirubin-PRE	0.632	9.25 µmol/L	60.0	64.8	1.70	0.015	NS	NS	NS
Lactate									
LAC-POST	0.681	2.8 mmol/L	53.3	81.2	2.83	<0.001	9.283	2.883–29.898	<0.001
LAC-POD1	0.768	2.4 mmol/L	63.3	76.2	2.66	0.001	NS	NS	NS
or:									
LAC-DIFF	0.710	−0.7 mmol/L	53.3	86.1	3.83	<0.001	5.518	1.953–15.590	0.001
Surgery									
OLR vs. LLR						0.015	NS	NS	NS
Extent of LR									
Minor vs. major LR						0.001	NS	NS	NS
or:									
EHH						0.002	NS	NS	NS
Complex resection						<0.001	4.144	1.310–13.106	0.016
Lymphadenectomy						0.014	NS	NS	NS
Duration PM	0.592	20.5 min	80.0	45.9	1.48	0.013	3.957	1.184–13.225	0.025
Blood loss	0.634	175 mL	96.4	24.5	1.28	0.037	NS	NS	NS
RBC substitution						0.013	NS	NS	NS
Operation time	0.688	312 min	76.7	56.6	1.77	0.005	NS	NS	NS

Area under the ROC curve (AUC) calculated cutoff values, sensitivity, specificity, and likelihood ratios and uni- and multivariate regression analysis of selected predictors of postoperative complications (Clavien–Dindo ≥ IIIb). The lactate-difference (LAC-DIFF) was calculated as follows: LAC-DIFF = LAC-POST − LAC-POD1. The following variables were tested in the univariate analysis but failed to show significance: Benign vs. malign disease, base excess before liver resection, body mass index, creatinine before liver resection, gender, and lactate before liver resection, parenchymal liver disease (steatosis, fibrosis or cirrhosis), use of T-tubes. ASA, American Society of Anesthesiology; AUC, area under the curve; DIFF, difference; EHH, extended hemihepatectomy; LR, liver resection; LLR, laparoscopic liver resection; LRa, likelihood ratio; NS, not significant; OLR, open liver resection; OR, odds ratio; PM, Pringle maneuver; POD1, postoperative day 1; POST, postoperative day 0; PRE, preoperative; RBC, red blood cells; ROC, receiver operator characteristic, Sen, Sensitivity; Spe, specificity; UVA, univariate analysis.

**Table 5 jcm-12-02100-t005:** Predictors of postoperative mortality in 152 patients undergoing liver resection.

Variables	Mortality								
	ROC					UVA	Multivariate Analysis		
	AUC	Cut-Off Value	Sen (%)	Spe (%)	LRa	*p*-Value	OR	95% CI	*p*-Value
Laboratory tests									
Bilirubin-PRE	0.827	13.85 µmol/L	71.4	76.6	4.32	0.003	NS	NS	NS
Lactate									
LAC-POST	0.800	3.1 mmol/L	71.4	80.0	3.57	0.009	11.685	1.757–77.730	0.011
LAC-POD1	0.838	5.4 mmol/L	71.4	94.5	12.95	0.021	NS	NS	NS
or:									
LAC-DIFF	0.822	−2.4 mmol/L	71.4	95.9	17.26	<0.001	20.724	2.722–157.812	0.003
Surgery									
Extent of LR									
Minor vs. major LR						0.017	NS	NS	NS
or:									
EHH						0.004	NS	NS	NS
Complex resection						0.001	39.925	4.032–375.805	0.002
Operation time	0.858	426 min	85.7	82.8	4.97	0.004	NS	NS	NS

Area under the ROC curve (AUC) calculated cutoff values, sensitivity, specificity, and likelihood ratios and uni- and multivariate regression analysis of selected predictors of postoperative mortality. Lactate-difference (LAC-DIFF) was calculated as follows: LAC-DIFF = LAC-POST − LAC-POD1. The following variables were tested in the univariate analysis but failed to show significance: ASA (American Society of Anesthesiology) score, base excess before liver resection, benign vs. malign disease, body mass index, creatinine before liver resection, duration of Pringle maneuver, gender, intraoperative blood loss, lactate before liver resection, lymphadenectomy, open vs. laparoscopic liver resection, patient age, parenchymal liver disease (steatosis, fibrosis or cirrhosis), the substitution of red blood cells, use of T-tubes. AUC, area under the curve; DIFF, difference; EHH, extended hemihepatectomy; LR, liver resection; LRa, likelihood ratio; NS, not significant; OR, odds ratio; POD1, postoperative day 1; POST, postoperative day 0; PRE, preoperative; ROC, receiver operator characteristic, Sen, Sensitivity; Spe, specificity; UVA, univariate analysis.

## Data Availability

Our database contains highly sensitive data that may provide insight into clinical and personnel information about our patients and lead to the identification of these patients. Therefore, according to organizational restrictions and regulations, these data cannot be made publicly available. However, the datasets used and/or analyzed during the current study are available from the corresponding author on reasonable request.

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
