# Peer review of "Prediction of the Postoperative Outcome in Liver Resection Using Perioperative Serum Lactate Levels"

_jcm, 2023, doi:10.3390/jcm12062100_

Round 1

Reviewer 1 Report

The authors describe prediction of the postoperative outcome in liver resection using 2 perioperative serum lactate levels. This manuscript is an interesting and acceptable result. This manuscript needs revision but may be accepted for publication.

1.     All seven deaths were cases of extended hemihepatectomy, and the mortality rate is high at 22.6% (7/31) when limited to cases of extended hemihepatectomy. Are there any technical or surgical indication issues? Were there any events during surgery that raised lactate levels, such as long Pringle's maneuver times? A discussion of mortality cases should be described in detail.

2.     The difference between liver failure and liver dysfunction is unclear. Describe the definition of liver failure.

Reviewer 2 Report

The authors present and interesting manuscript on the relation of serum lactate levels and postoperative morbidity and mortality after hepatic resections. The paper is well disigned and performed. The results obtained support the monitorization of lactate levels in patients undergoing these procedures. 

Only one question. Did the authors have done some technique to calculate the volumen of the hepatic remanent after resection? Did they performe some technique to obtain hypertrophy the hepatic remanent ? May be a relationship between amaunt of liver remanent, postoperative hepatic insufficiency and lactate levels?  
